# Molecular Basis of Multiple Mitochondrial Dysfunctions Syndrome 2 Caused by CYS59TYR BOLA3 Mutation

**DOI:** 10.3390/ijms22094848

**Published:** 2021-05-03

**Authors:** Giovanni Saudino, Dafne Suraci, Veronica Nasta, Simone Ciofi-Baffoni, Lucia Banci

**Affiliations:** 1Magnetic Resonance Center (CERM), University of Florence, 50019 Sesto Fiorentino, Italy; saudino@cerm.unifi.it (G.S.); dafne.suraci@gmail.com (D.S.); nasta.veronica@gmail.com (V.N.); 2Department of Chemistry “Ugo Schiff”, University of Florence, 50019 Sesto Fiorentino, Italy; 3Consorzio Interuniversitario Risonanze Magnetiche di Metalloproteine (CIRMMP), 50019 Sesto Fiorentino, Italy

**Keywords:** BOLA3, GLRX5, multiple mitochondrial dysfunctions syndrome, iron–sulfur protein, mitochondria, MMDS2

## Abstract

Multiple mitochondrial dysfunctions syndrome (MMDS) is a rare neurodegenerative disorder associated with mutations in genes with a vital role in the biogenesis of mitochondrial [4Fe–4S] proteins. Mutations in one of these genes encoding for BOLA3 protein lead to MMDS type 2 (MMDS2). Recently, a novel phenotype for MMDS2 with complete clinical recovery was observed in a patient containing a novel variant (c.176G > A, p.Cys59Tyr) in compound heterozygosity. In this work, we aimed to rationalize this unique phenotype observed in MMDS2. To do so, we first investigated the structural impact of the Cys59Tyr mutation on BOLA3 by NMR, and then we analyzed how the mutation affects both the formation of a hetero-complex between BOLA3 and its protein partner GLRX5 and the iron–sulfur cluster-binding properties of the hetero-complex by various spectroscopic techniques and by experimentally driven molecular docking. We show that (1) the mutation structurally perturbed the iron–sulfur cluster-binding region of BOLA3, but without abolishing [2Fe–2S]^2+^ cluster-binding on the hetero-complex; (2) tyrosine 59 did not replace cysteine 59 as iron–sulfur cluster ligand; and (3) the mutation promoted the formation of an aberrant apo C59Y BOLA3–GLRX5 complex. All these aspects allowed us to rationalize the unique phenotype observed in MMDS2 caused by Cys59Tyr mutation.

## 1. Introduction

Multiple mitochondrial dysfunctions syndrome (MMDS) is a rare, severe autosomal recessive disorder of the energy metabolism with onset in early infancy, characterized by markedly impaired neurological development, weakness, respiratory failure, lactic acidosis, hyperglycinemia, and early fatality [1,2,3,4]. Mutations in genes encoding for NFU1, BOLA3, IBA57, ISCA2, and ISCA1 proteins lead to MMDS types 1 to 5, respectively. All these five genes play an essential role in the biogenesis of mitochondrial [4Fe–4S] cluster-binding proteins [5,6,7,8,9,10], which, in humans, consist of the respiratory chain complexes I and II, aconitase, the lipoic acid synthase, the molybdenum cofactor biosynthesis protein 1, and the electron transfer flavoprotein-ubiquinone oxidoreductase involved in the β-oxidation of lipids [2]. Thus, MMDS types 1–5 induce impairment of cellular respiration and lipoic acid metabolism [5,6,8].

Bi-allelic variants in BOLA3 cause MMDS type 2 with hyperglycinaemia (MMDS2; MIM#614299), typically characterized by infantile encephalopathy, leukodystrophy, lactic acidosis, non-ketotic hyperglycinemia and death in early childhood [5,7,11]. Six missense and six non-sense disease-causing variants in BOLA3 have been identified to date in patients affected by MMDS2 [5,7,11,12,13,14,15,16,17,18,19,20,21]. Recently, a novel phenotype for MMDS2 with complete clinical recovery and partial resolution of magnetic resonance imaging abnormality was observed in a patient [18]. Whole genome sequencing on the latter MMDS2 patient identified compound heterozygous variants in BOLA3: one previously reported (c.136C > T, p.Arg46*, paternally inherited [7,12]) and one novel variant (c.176G > A, p.Cys59Tyr, maternally inherited). These heterozygous variants determine a much milder phenotype with respect to homozygous c.136C > T (p.Arg46*) variants reported by Baker et al. in three unrelated patients [7]. The essential discrepancy is the different life span: while the phenotypes at 18 months are comparable, the grown patient (after 8 years old) regained normal neurological and cognitive function until a complete clinical recovery only in the heterozygous BOLA3 variants.

Cysteine-59 in BOLA3 was shown to be a coordinating ligand of a [2Fe–2S]^2+^ cluster bound to a hetero-dimeric complex formed by BOLA3 and its protein partner GLRX5 [22]. Specifically, the [2Fe–2S]^2+^ cluster is bridged between the two proteins being coordinated, from the GLRX5 side, by Cys67 and by the cysteine of a GLRX5-bound glutathione (GSH) molecule, and, from the BOLA3 side, by Cys59 and His96. A combination of in vitro and in vivo studies support that BOLA3 complexed with GLRX5 could function in [2Fe–2S] cluster trafficking and/or insertion reactions in the mitochondrial iron–sulfur protein biogenesis [23]. Recently, the BOLA3–GLRX5 complex in its [2Fe–2S]^2+^ bound state was shown to transfer the cluster to both apo human ferredoxins FDX1 and FDX2 with rate constants comparable to other cluster donors to FDX proteins [24,25], as well as to transfer its cluster to apo NFU1 to form a [4Fe–4S]^2+^ cluster-bound NFU1 dimer [26]. However, considering that the yeast homologue of human BOLA3 was shown not to be required for the maturation of mitochondrial [2Fe–2S] proteins [23], the cluster transfer to FDXs is very likely not physiologically relevant. On the contrary, the BOLA3-driven cluster assembly on NFU1 is supported by in vivo studies. NFU1 is the protein of the mitochondrial iron-sulfur cluster (ISC) assembly machinery implicated in inserting the [4Fe–4S]^2+^ cluster into specific mitochondrial client proteins, i.e., lipoic acid synthase and components of respiratory complexes I and II [6,27,28]. Recently, a molecular view of how NFU1 cooperates with ISCA1 and ISCA2 proteins in the maturation of mitochondrial [4Fe–4S] proteins has been defined [29]. MMDS type 1 patients harboring a frame-shift mutation in the NFU1 gene presented with a biochemical phenotype similar to BOLA3-deficient individuals affected by MMDS2 [30,31,32], suggesting that the two proteins take part to the same cellular pathway, in agreement with the in vitro BOLA3–GLRX5–NFU1-dependent pathway previously reported [26]. Following this molecular picture, the impairment of the latter [4Fe–4S] protein assembly pathway, caused by the pathogenic mutations found in MMDS1 and MMDS2, might explain the cellular functional defects associated to the lack of an efficient biogenesis of lipoic acid synthase and components of respiratory complexes I and II [5]. However, BOLA3 and NFU1 were also implicated at another stage of the mitochondrial ISC machinery that does not involve GLRX5. Indeed, BOLA3 was proposed to drive the transfer of the [4Fe–4S]^2+^ cluster bound to NFU1 into specific mitochondrial target proteins [28]. On the other hand, the interaction between NFU1 and BOLA3 was not clearly detected in vivo [28], and recent in vitro data showed that BOLA3 and NFU1 do not interact each other in both apo and holo forms [24,26]. Thus, these in vitro data suggest that BOLA3 might be not involved at this stage of the mitochondrial ISC assembly machinery, but only when it works in the complex with GLRX5.

The aim of this work was to unravel the molecular basis of the Cys59Tyr BOLA3 pathogenic mutation in MMDS2 in order to comprehend its novel and unique phenotype never documented in MMDS, i.e., associated with a complete clinical recovery. To do so, we first investigated the structural impact of the Cys59Tyr mutation on BOLA3 by NMR, and then we analyzed how the mutation impacts on BOLA3–GLRX5 complex formation and on the hetero-complex cluster-binding properties by various spectroscopic techniques, including NMR, UV–VIS circular dichroism, UV–VIS and fluorescence spectroscopies, as well as by experimentally driven molecular docking. We show that 1) the mutation does not modify the overall structure of BOLA3, but it structurally perturbs the iron–sulfur (Fe–S) cluster-binding region; 2) the mutation promotes the formation of an aberrant apo C59Y BOLA3–GLRX5 complex structurally different from that formed by the wild-type proteins; 3) although the mutation does not abolish [2Fe–2S]^2+^ cluster-binding on the hetero-complex, it determines a perturbation in the chemical environment of the [2Fe–2S]^2+^ cluster due to the lack of the Cys59 Fe–S cluster ligand, which it is not replaced by the tyrosine. All these aspects allowed us to rationalize the unique phenotype observed for the Cys59Tyr BOLA3 pathogenic mutation in MMDS2.

## 2. Results

### 2.1. Cys59Tyr Mutation Structurally Perturbed Only the Fe–S Cluster-Binding Region of BOLA3

The pathogenic mutation of cysteine-59 to a tyrosine did not drastically affect the structural properties of BOLA3 protein. The ^1^H–^15^N HSQC spectrum of ^15^N labelled Cys59Tyr BOLA3 mutant (C59Y BOLA3, hereafter) showed, indeed, a well-folded protein, in agreement with all the cross-peaks having wide dispersions and sharp linewidths (Figure 1A). The ^1^H–^15^N HSQC spectrum of ^15^N-labelled C59Y BOLA3 experienced some differences when compared to that of the wild-type protein, but the same protein fold was maintained (Figure 1A). The backbone chemical shift differences between wild-type and C59Y BOLA3 proteins indicated that the Cys59Tyr mutation in BOLA3 affected some residues of the loop containing the mutation and the spatially close, fully conserved His96 (Figure 1B), which is the other [2Fe–2S] cluster ligand of BOLA3 in the BOLA3–GLRX5 hetero-dimeric complex [22]. When, in a previous study [22], Cys59 was mutated to Ala, the residues with significant backbone chemical shift differences were similar (when calculated applying the same chemical shift threshold value of 0.06 on both C59A and C59Y BOLA3 mutants). The only differences consisted of two less residues perturbed in the loop containing the mutation, once Ala 59 was present, and of the loop containing the His96 ligand that was not perturbed by the C59A mutation. This indicates that the introduction of a tyrosine with respect to an alanine perturbed slightly more the local structural environment of the cluster-binding region of BOLA3, in agreement with the bulky steric hindrance of the tyrosine vs. that of the alanine that has a steric hindrance similar to that of the cysteine.

In conclusion, the NMR data showed that the Cys59Tyr mutation did not affect the protein fold, but that only the Fe–S cluster-binding region resulted in being structurally perturbed.

### 2.2. Cys59Tyr BOLA3 Mutation Modified the Interaction of BOLA3 with Apo GLRX5

BOLA3 is known to form a hetero-dimeric complex with apo GLRX5 with an affinity constant of 1.2 × 10^5^ M^−1^ [22,23,24,25]. To test whether the Cys59Tyr mutation affected the formation of the apo BOLA3–GLRX5 complex, we stepwise added apo GLRX5 to ^15^N-labelled C59Y BOLA3, or vice versa adding C59Y BOLA3 to apo ^15^N-labelled GLRX5, and the interaction was followed through ^1^H−^15^N HSQC NMR experiments (see the Materials and Methods section for details). Spectral changes occurred on both titrations, and the NMR signals of the free and the bound proteins were in fast and intermediate exchange regimes relative to the NMR time scale (Figure 2). These data indicate that the Cys59Tyr mutation did not impair apo hetero-complex formation. Moreover, the NMR titration data (see the Materials and Methods section for details) showed that the two proteins interacted both in the absence and in the presence of GSH, and that GSH, which bound to the GLRX5 subunit of the hetero-complex, was involved at the protein–protein interface, as it occurred for the wild-type protein.

As following step, chemical shift perturbations and line broadening analyses were performed by comparing the ^1^H−^15^N HSQC spectra of ^15^N-labelled BOLA3 and ^15^N-labelled C59Y BOLA3, respectively, with those recorded in presence of equimolar amounts of apo GLRX5, at two different starting concentrations of ^15^N-labelled BOLA3 and ^15^N-labelled C59Y BOLA3 (100 and 800 μM). The observed spectral changes were then mapped on the solution structures of BOLA3 (PDB 2NCL) and on a structural model of C59Y BOLA3, obtained by homology modelling through MODELLER 9.10 software followed by energy minimization in water through AMBER 12.0 program (see the Materials and Methods section for details).

In the titrations performed at 100 μM protein concentration, the spectral analyses showed that very similar interacting regions were identified on both BOLA3 proteins, i.e., wild-type and Cys59Tyr mutant (Figure 3A,B, respectively). These regions included helix α_2_; strand β_3_; the loop connecting the latter two elements, which contains His96 Fe–S cluster ligand; and a few residues following the Cys59 Fe–S cluster ligand (Figure 3A,B). Comparing the chemical shifts of the backbone NH signals that were in fast exchange on the NMR time scale on the ^1^H−^15^N HSQC spectra of ^15^N-labelled BOLA3 and ^15^N-labelled C59Y BOLA3 acquired at the final 1:1 ratio, we found that they had the same values, suggesting that the hetero-complex was formed in both cases at the same level and thus that the Cys59Tyr mutation of BOLA3 did not significantly affect the affinity relative to hetero-complex formation with apo GLRX5. This was confirmed by an NMR titration, where a 1:1 ^15^N-labelled BOLA3-^15^N-labelled C59Y BOLA3 mixture was titrated with unlabeled apo GLRX5. Indeed, in the latter experiment, we were able to follow the stepwise interaction of apo GLRX5 concomitantly occurring with both ^15^N-labelled BOLA3 and ^15^N-labelled C59Y BOLA3, mixed at a 1:1 ratio. We found that apo GLRX5 interacted at the same extent with both BOLA3 proteins from substoichiometric ratios up to when the final 2:1 ratio was reached, indicating that the added apo GLRX5 was equally partitioned between the two BOLA3 proteins along the additions (Appendix A).

The spectral analysis performed at 800 μM starting protein concentration also showed that, in addition to the interacting regions mentioned above in the titrations performed at 100 μM protein concentration, the loop of BOLA3 containing Cys59Tyr mutation was also affected in the C59Y BOLA3–GLRX5 interaction, experiencing all its backbone NHs line broadening beyond detection, while the same loop was not involved in the wild-type BOLA3–GLRX5 interaction (Figure 3C,D). This result indicates that the Cys59Tyr mutation can promote the formation of an aberrant interaction in the hetero-complex characterized by protein–protein contacts that involve the loop region containing the Cys59Tyr mutation and that are not present in the wild-type protein–protein interaction.

Spectral changes were also analyzed on the ^1^H−^15^N HSQC map of ^15^N-labelled apo GLRX5 at a starting protein concentration of 800 μM upon the interaction with BOLA3 or C59Y BOLA3 and mapped on the solution structure of GSH-bound GLRX5 (PDB 2WUL) (Figure 4). Similar interaction regions were observed in both wild-type and C59Y BOLA3–GLRX5 titrations. They involved the residues in contact with GSH and with the Cys67 Fe–S cluster ligand of GLRX5, indicating that GLRX5 interacted with both BOLA3 and C59Y BOLA3 proteins via the same Fe–S cluster binding site region (Figure 4). However, a higher number of signals of GLRX5, which specifically display line broadening beyond signal detection, were observed in the Cys59Tyr BOLA3 mutant interaction with respect to the wild-type BOLA3 interaction (Figure 4), similarly to what was observed on the BOLA3 side (Figure 3C,D). 

In order to structurally visualize the aberrant protein–protein interaction induced by the Cys59Tyr mutation, we carried out a protein–protein docking modeling to obtain the structural models of apo BOLA3–GLRX5 and apo C59Y BOLA3–GLRX5 complexes, using, as experimental data, the chemical shift perturbations observed on C59Y BOLA3, BOLA3, and apo GLRX5 proteins for the NMR titrations performed at 800 μM starting protein concentration. Structural models of apo BOLA3–GLRX5 and apo C59Y BOLA3–GLRX5 complexes were calculated following the standard protocol of HADDOCK 2.2 docking program [33,34] and considering the GSH Fe–S cluster ligand bound to GLRX5, according to what has been previously reported [22] (see the Materials and Methods section for details). The docking models were clustered on the basis of common contacts following the standard HADDOCK scoring approach [35] (Appendix A). From the comparison of the two best models, we found that once the GLRX5 structure in the two hetero-complexes was superimposed, it appeared that C59Y BOLA3 had a different orientation with respect to BOLA3, with the latter hetero-complex being more compact than that with C59Y BOLA3 (Figure 5). Specifically, the loop between the β1 and β2 strands, where Tyr59 of C59Y BOLA3 was located, became closer to the [2Fe–2S] cluster ligands of GLRX5 (Cys67 and GSH), with respect to what occurred for Cys59 of BOLA3 in the BOLA3–GLRX5 complex. Overall, these NMR-based docking models showed that the Cys to Tyr mutation triggered a close interaction between the C59Y-mutated loop and the Fe–S cluster ligands of GLRX5. This turned out in a significant alteration of the structural arrangement in the apo C59Y BOLA3–GLRX5 complex, which is, indeed, less compact and more elongated with respect to that of the wild-type hetero-complex.

In conclusion, the NMR data indicate that the Cys59Tyr mutation did not abolish the formation of an apo hetero-complex between BOLA3 and GLRX5, but it significantly modified the interaction between the two proteins, promoting the formation of a hetero-complex structurally different from that formed by the wild-type protein.

### 2.3. C59Y BOLA3–GLRX5 Complex Bound a [2Fe–2S]^2+^ Cluster

In light of the different interaction properties of the C59Y BOLA3 mutant with apo GLRX5 in comparison with those of wild-type BOLA3, we investigated whether the C59Y BOLA3–GLRX5 complex is still able to bind a [2Fe–2S]^2+^ cluster, as the BOLA3–GLRX5 complex does [22,23]. To address this question, we chemically reconstituted the apo C59Y BOLA3–GLRX5 complex, obtained by mixing equimolar amounts of apo GLRX5 and C59Y BOLA3, and then analyzed the chemically reconstituted mixture by UV–VIS, UV–VIS circular dichroism (CD), and paramagnetic 1D ^1^H NMR spectroscopies. 

The data showed that the C59Y BOLA3–GLRX5 complex was still able to bind a [2Fe–2S]^2+^ cluster: indeed, the UV–VIS spectrum had absorption bands between 300 and 600 nm, typical of biological oxidized [2Fe–2S]^2+^ centers [36,37] (Figure 6A); the UV–VIS CD spectrum showed well-structured, intense bands typical of protein-bound [2Fe–2S] cluster [38,39] (Figure 6B); and paramagnetic 1D ^1^H NMR spectrum showed broad signals in the 35–20 ppm range and of a sharper one at 13 ppm, all of them typical of residues bound to an oxidized [2Fe–2S]^2+^ cluster [40,41,42] (Figure 6C).

Comparing the UV–VIS, UV–VIS CD, and paramagnetic 1D ^1^H NMR spectra of the [2Fe–2S]^2+^ C59Y BOLA3–GLRX5 complex with the corresponding spectra obtained on the [2Fe–2S]^2+^ BOLA3–GLRX5 complex, we observed significant differences, indicating that the Cys-to-Tyr mutation perturbed the cluster-binding environment. Specifically, the absorption bands at 398, 440, 512, and 593 nm in the [2Fe–2S]^2+^ BOLA3–GLRX5 complex were shifted to 407, 459, 507, and 540 nm in the [2Fe–2S]^2+^ C59Y BOLA3–GLRX5 (Figure 6A). Considering that it has previously been shown that the UV–VIS absorption spectra of [2Fe–2S]^2+^ ferredoxins display hypsochromic shifts in the 400–500 nm region upon the replacement of a sulfur by an oxygen ligand [43,44], the bathochromic shifts observed in the UV–VIS spectrum of the C59Y BOLA3–GLRX5 complex, as compared to the spectrum of the wild-type hetero-complex, suggests that the tyrosine did not coordinate the cluster. This is consistent with the fact that the coordination of a Tyr to an Fe–S cluster is extremely rare and previously detected only for a [4Fe–4S] cluster of the hydrogenase maturase HydE from *Thermotoga maritima* and for the P-cluster in *G. diazotrophicus* nitrogenase [45,46], but never for a [2Fe–2S] cluster. Moreover, the UV–VIS CD and paramagnetic 1D ^1^H NMR spectra indicated that the Cys-to-Tyr mutation affected the cluster-binding environment. Indeed, the UV–VIS CD spectrum of the [2Fe–2S]^2+^ C59Y BOLA3–GLRX5 complex differed from that of the [2Fe–2S]^2+^ BOLA3–GLRX5 complex, showing an opposite trend of the intensities of the two bands in the region 350–425 nm (Figure 6B), as well as the hyperfine-shifted NMR signals, which are typical of βCH_2_ and αCH protons of the residues bound to a [2Fe–2S]^2+^ cluster, show changes in the chemical shift values when comparing the paramagnetic 1D ^1^H NMR spectra of the two [2Fe–2S]^2+^ hetero-complexes (Figure 6C).

In order to further examine the potential involvement of Tyr59 in cluster-binding in place of Cys59, we performed fluorescence emission spectroscopy, which has been extensively applied to discriminate between a tyrosine having a protonated or a deprotonated phenolic side-chain group (i.e., phenol- or phenoxide-tyrosine hereafter, respectively) in proteins [47,48,49,50], with the phenoxide state of the tyrosine being the only one able to coordinate a metal cofactor. A phenoxide-tyrosine has an emission maximum at 345 nm [51] while a phenol-tyrosine has an emission maximum at 303 nm [52]. Fluorescence emission spectra of native proteins containing Trp, Tyr, and Phe residues (as it occurs in the BOLA3–GLRX5 complex) typically show an emission maximum in the range of 331 to 343 nm [53] as a consequence of the predominant contribution of fluorescence emission of Trp [54] over that of Tyr and Phe residues [55]. The fluorescence emission spectrum of apo C59Y BOLA3–GLRX5 complex shows a band with a maximum at 335 nm (Figure 7) due to the presence of a Trp in GLRX5, while the fluorescence emission spectrum of [2Fe–2S]^2+^ C59Y BOLA3–GLRX5 is significantly different, showing, in addition to the band at 335 nm due to the Trp, an intense band at 308 nm in the fluorescence region typical of a phenol-tyrosine (Figure 7). On the contrary, no apparent changes were observed in the fluorescence region typical of a phenoxide-tyrosine at 345 nm. This is better visualized by the difference of the two fluorescence emission spectra (Figure 7, inset), which resulted in a positive band with a maximum at 303 nm, typical of a phenol-tyrosine, and with no positive band with a maximum centered at 345 nm, typical of a phenoxide-tyrosine. Therefore, the fluorescence emission data strongly support the idea that Tyr59 is not involved in cluster coordination. A possible alternative Fe–S cluster ligand in place of Cys59 might be the cysteine of a GSH molecule additionally bound to the hetero-complex. However, it is highly unlikely considering that we previously showed that the C59A mutation, which provides structural perturbations on BOLA3 quite similar to those determined by the C59Y mutation (see Section 2.1), do not promote the binding of a second GSH molecule to the [2Fe–2S] cluster bound to the hetero-complex [22].

Finally, considering that we have recently shown the transfer of [2Fe–2S] clusters bound to BOLA3–GLRX5 to apo NFU1 to form a [4Fe–4S]^2+^ cluster-bound NFU1 dimer [26], we investigated by NMR the effect of the Cys59Tyr mutation on this process. Specifically, by replicating on the Cys59Tyr mutant the ^1^H−^15^N HSQC NMR experiments previously performed on the wild-type protein [26] (see the Materials and Methods section for details), we found that, although the [4Fe–4S] cluster assembly on NFU1 was not abolished, a significant lower efficiency was observed for the mutant with respect to the wild-type protein. Specifically, the amount of [4Fe–4S] NFU1 that was formed in the final mixture was 25% (C59Y mutant) vs. 65% (wild-type).

In conclusion, all the observed spectroscopic data were consistent with the presence of a [2Fe–2S]^2+^ cluster bound to the C59Y BOLA3–GLRX5 complex, as it occurred in the wild-type hetero-complex. However, they also showed that (1)the Cys59Tyr mutation determined a perturbation in the chemical environment of the [2Fe–2S]^2+^ cluster due to the absence of the Cys59 Fe–S cluster ligand, (2) Tyr 59 did not replace Cys59 as Fe–S cluster ligand, and (3) the Cys59Tyr mutation reduced the efficiency of [4Fe–4S] cluster assembly on NFU1.

## 3. Discussion

BOLA3 is a protein required in human cells for the assembly of mitochondrial [4Fe–4S] cluster-binding target proteins that are involved in central pathways of mitochondrial energy production and carbon metabolism [5,11,23,28]. As a result of this, the impairment of BOLA3 function causes severe cellular defects, which are associated with a mitochondrial human disease identified as MMDS2 [1,5,7,15]. The number of disease-causing variants found in BOLA3 has recently increased and, among them, a novel phenotype for MMDS2, with complete clinical recovery after 8 years old, was observed in a patient [18]. This patient was characterized by heterozygous variants in BOLA3, i.e., c.136C > T, p.Arg46* (paternally inherited) and c.176G > A, p.Cys59Tyr (maternally inherited). The peculiarity of these heterozygous variants is that they determine a much milder phenotype with respect to homozygous c.136C > T (p.Arg46*) variants reported by Baker et al. in three unrelated patients [7]. The latter variant resulted, indeed, incompatible with life, due to the fact that the absence of BOLA3 cannot be compensated by other systems in the metabolic cell request. This suggested us that Cys59Tyr BOLA3 variant might, on the contrary, maintain some functional activity and thus might compensate, at least partially, the effect of the other p.Arg46* BOLA3 heterozygous variant. In order to characterize at the molecular level this novel and unique heterozygous phenotype never documented in MMDS2, we here investigated the impact of the Cys59Tyr mutation on BOLA3 structure, on the hetero-complex formation between BOLA3 and its partner protein GLRX5 and on the Fe–S cluster binding and transfer properties of the hetero-complex.

We showed that the Cys59Tyr mutation did not have severe effects on the overall protein fold of BOLA3. C59Y BOLA3 resulted indeed as a well-structured protein with the same fold of the wild-type protein. These data suggest that the pathogenic effects of the mutation did not promote increased folding instability and thus the protein turnover in cells is expected to not be altered with respect to that of the wild-type protein. We also found that the mutation did not impair apo protein–protein interaction between BOLA3 and GLRX5 and did not significantly affect the affinity constant for the hetero-complex formation. C59Y BOLA3 indeed maintained the same ability to interact with GLRX5 as wild-type BOLA3. However, NMR protein–protein interaction data showed that tyrosine 59 promoted the formation of an aberrant interaction with GLRX5, which was not present in wild-type BOLA3. This effect can be structurally rationalized considering both the high solvent exposure of the mutated position and that the mutation did not dramatically modify the protein surface between the two proteins. Indeed, Cys59 is highly solvent-exposed, being located in at the edge of an extended loop of BOLA3 [23]. When substituted by a tyrosine, the latter residue, which remained still solvent-exposed in the C59Y BOLA3 structure (Appendix A), had the ability, thanks to its aromatic ring, to trigger, in the apo C59Y BOLA3–GLRX5 complex, an intermolecular protein–protein interaction not occurring in the wild-type hetero-complex. The structural model of the apo C59Y BOLA3–GLRX5 complex showed, indeed, that Tyr59 closely interacted with the glutathione molecule bound to the hetero-complex as well as with the Cys67 Fe–S cluster ligand of GLRX5, causing a structural conformation of the apo hetero-complex different from that observed when Cys59 was present. The formation of this aberrant hetero-complex might represent the cause of the pathogenic effects induced by the Cys59Tyr mutation. NMR data also showed that the formation of this Tyr59-driven interaction depended on protein concentration, thus indicating the presence of an equilibrium between a native-like hetero-complex conformation and an aberrant hetero-complex conformation. Specifically, the NMR data support that this equilibrium occurred involving multiple binding states of comparable energy. Indeed, the majority of the backbone NHs of the residues at the protein–protein interface of the apo C59Y BOLA3–GLRX5 complex showed line broadening beyond detection on both interacting proteins that was not recovered by adding an excess of the protein titrant, as it would be expected in the case of a single-mode binding [56]. This effect was accompanied by a higher number of signals being involved at the interacting interfaces of the two proteins, when the Cys59Tyr mutation was present (Figure 3B,D and Figure 4), suggesting that the mutation had the following effect: the interacting proteins in their associated state sampled each other’s surfaces, increasing the number of contacts to find optimal binding geometry, which is, however, prevented by the mutation, thus mimicking a condition commonly known as encounter complex [57]. Overall, our data suggested that the Cys59Tyr mutation does not impair the formation of a native-like hetero-complex, but it can substantially modify it. This model allows the “reversible” phenotype observed in the patient affected by the p.Arg46*/Cys59Tyr heterozygous variants to be explained [18]. We can suggest, indeed, that the Cys59Tyr mutation does not completely abolish the physiological function performed by wild-type BOLA3–GLRX5 complex, in such a way partially compensating the complete lack of function of the other p.Arg46* BOLA3 heterozygous variant [7]. In the early human development phases during which the functional processes performed by BOLA3 are much more critical, the performance of the native-like hetero-complex formed by C59Y BOLA3 might not be sufficient to sustain the correct human development in that period of life, as the aberrant non-functional complex is also formed. Once a less metabolic demand of the hetero-complex is needed after the early stages of human development, such as it occurs at the age of eight years, a complete clinical recovery can occur as the levels of native-like hetero-complex formed by C59Y BOLA3 can fully satisfy the BOLA3-dependent physiological processes.

It has been proposed, on the basis of a combination of in vitro and in vivo studies [23,28], that the BOLA3–GLRX5 complex transfers its [2Fe–2S]^2+^ cluster to protein partners, such as NFU1 [26], which are involved in the final steps of the maturation of mitochondrial [4Fe–4S] cluster-binding target proteins [28,58]. On this respect, it is also relevant to investigate how the Cys59Tyr mutation affects the Fe–S cluster-binding properties of the hetero-complex, as Cys59 is a ligand of the [2Fe–2S]^2+^ cluster in the hetero-complex [22]. Spectroscopic data showed that the C59Y BOLA3–GLRX5 complex maintained its ability to bind a [2Fe–2S]^2+^ cluster even if the tyrosine did not substitute the cysteine in [2Fe–2S] cluster-binding. Thus, the Cys59Tyr mutation, in addition of altering the apo BOLA3–GLRX5 complex conformation, modified the coordination mode of the cluster with respect to the wild-type hetero-complex. This aspect might also play a key role in the pathogenic effects of the Cys59Tyr mutation. Specifically, the ability of the BOLA3–GLRX5 complex in transferring its [2Fe–2S]^2+^ cluster to protein partners might be altered in the Cys59Tyr mutant, as well as the cluster stability towards its degradation in the cellular environment might be affected. In agreement with this view, NMR data showed a lower efficiency of [4Fe–4S] cluster assembly on NFU1 by [2Fe–2S] C59Y BOLA3–GLRX5 with respect to the wild-type hetero-complex, suggesting that the BOLA3–GLRX5–NFU1 cellular pathway might be negatively perturbed by the Cys59Tyr mutation. The fact that [2Fe–2S] cluster-binding is not abolished by the Cys59Tyr mutation, but it is only modified, again agrees with the MMDS2 phenotype: indeed, the modified coordination properties of the hetero-complex might reduce the BOLA3 function in a period of the human body development with a high BOLA3-dependent metabolic demand. However, the ability of the hetero-complex to still coordinate the cluster guarantees a certain level of cellular functionality of the hetero-complex, which can thus rationalize the observed complete clinical recovery when the BOLA3-dependent metabolic requirements are less demanding.

In conclusion, our in vitro study investigating the effects of the Cys59Tyr mutation on the structural, protein–protein interaction, and cluster-binding and -transfer properties of BOLA3/BOLA3–GLRX5 complex sheds light on the molecular grounds of the Cys59Tyr variant-dependent MMDS2, explaining its novel phenotype associated with a complete clinical recovery.

## 4. Materials and Methods

### 4.1. Protein Expression and Purification

Site-directed mutagenesis (QuickChange Site-directed Mutagenesis Kit, Agilent Technologies, Milan, Italy) was applied on the pETG20A/wild-type BOLA3 expression vector, already available in the lab [22], to produce recombinant C59Y BOLA3 mutant.

*Escherichia coli* BL21(DE3)-Gold (Agilent Technologies, Milan, Italy) competent cells were transformed with pETG20A plasmid containing N-terminal-tagged (N-terminal TRX-6His-tag) C59Y BOLA3. Cells were cultivated at 37 °C in 1 L of Luria–Bertani (LB) media adding ampicillin (100 μg/mL) until the OD_600_ reached 0.6–0.8. The protein expression was induced by adding 0.5 mM of isopropyl β-D-1-thiogalactopyranoside and shaking for 5 h at 30 °C, 200 rpm. The cells were harvested by centrifugation at 5000 rpm for 20 min (JA-10, Beckman Coulter, Milan, Italy). The cell pellet was resuspended in the binding buffer (50 mM phosphate buffer, 300 mM NaCl, 20 mM imidazole (pH 8.0)), and the cells were lysed by sonication (30 min, 2′’ ON and 9.9′’ OFF). The N-terminal TRX-6His-tag C59Y BOLA3 protein was purified from the lysate using a HisTrap HP column (GE Healthcare, Milan, Italy). The TRX-6His-tag was cleaved by tobacco etch virus protease overnight at room temperature in 50 mM phosphate buffer, 300 mM NaCl, and 20 mM imidazole (pH 8.0). HisTrap HP column was then performed to separate C59Y BOLA3 cleaved by the TRX-6His-tag from uncleaved TRX-6His-tag C59Y BOLA3. The final yield of C59Y BOLA3 was ≈40 mg per liter of LB culture. Recovered C59Y BOLA3 was pure enough to be used for spectroscopic and biochemical studies. All the expression and purification steps were performed in aerobic conditions. ^15^N-labelled C59Y BOLA3 was produced in M9 minimal medium supplemented with ^15^NH_4_Cl following the same procedure described for *E. coli* cells grown in LB medium. The expression and purification of wild-type BOLA3, apo GLRX5, and apo NFU1 were obtained as previously described [23,26,59]. 

### 4.2. In Vitro Chemical Reconstitution of the Hetero-Complexes

The apo BOLA3–GLRX5 and apo C59Y BOLA3–GLRX5 hetero-complexes were chemically reconstituted in anaerobic conditions in 50 mM Tris-HCl, 100 mM NaCl, and 5 mM DTT buffer at pH 8.0 with eightfold FeCl_3_ and Na_2_S for 16 h at room temperature. Fe–S cluster chemical reconstitution was performed with hetero-complex concentrations of ≈40–80 μM. Excess of FeCl_3_ and Na_2_S was anaerobically removed by passing the mixture on a PD-10 desalting column, and the holo hetero-complex was recovered. Anaerobic conditions were obtained performing the chemical reconstitution in an anaerobic chamber (O_2_ < 1 ppm) and by using all buffers degassed. 

### 4.3. NMR Spectroscopy to Characterize Proteins and Hetero-Complexes in Their Apo Forms

Diamagnetic NMR experiments were acquired at 298 K in 150 mM NaCl, 5 mM GSH, 5 mM DTT, 50 mM phosphate buffer (pH 7.0), and 10% (*v*/*v*) D_2_O. Diamagnetic NMR spectra were recorded on Bruker AVANCE 700 and 950 MHz spectrometers, processed using the standard Bruker software (Topspin), and analyzed with CARA program. Backbone weighted chemical shift differences (Δδ_avg_(HN)) were calculated by the equation Δδ_avg_(HN) = (((ΔH)^2^ + (ΔN/5)^2^)/2)^1/2^. The estimation of a chemical shift threshold value to define meaningful chemical shift differences was obtained by averaging Δδ_avg_(HN) values plus 1σ.

The formation of the apo C59Y BOLA3–GLRX5 complex was monitored by running ^1^H–^15^N HSQC NMR experiments. ^15^N-labelled C59Y BOLA3 at 100 or 800 μM starting protein concentration or ^15^N-labelled apo GLRX5 at 800 μM starting protein concentration was titrated with increasing amounts of unlabeled partner until the 1:2 protein ratio was reached. Spectral changes were monitored upon the addition of increasing amounts of the unlabeled partner. NMR data were analyzed following standard procedures [60,61]. The same set of NMR data were collected to monitor the formation of the apo BOLA3–GLRX5 complex. A 1:1 ^15^N-labelled BOLA3-^15^N-labelled C59Y BOLA3 mixture was also titrated with unlabeled apo GLRX5 up to a 1:1:2 ^15^N-labelled BOLA3-^15^N-labelled C59Y BOLA3–GLRX5 ratio to compare the relative affinity of GLRX5 versus the two BOLA3 proteins.

The experimental procedure followed in each NMR titration and the relative data analysis are reported hereinafter. In a first step, the ^15^N-labelled protein was titrated with the unlabeled protein partner in the absence of GSH but in the presence of 5 mM DTT, so as to avoid oxidative processes potentially occurring during the titration experiment. This allowed for exclusively following the effects of the protein–protein interaction on the backbone chemical shifts, avoiding chemical shift perturbations due to GSH–protein interaction. Indeed, while GSH does not interact with both wild-type and C59Y BOLA3 (Appendix A), GSH additions at millimolar concentration largely affects the chemical shifts of backbone NHs of apo GLRX5 [59], indicating a specific binding of GSH for GLRX5 and not for BOLA3. Once the end of the titration was reached, i.e., when no further chemical shifts were observed upon titrant additions indicating full hetero-complex formation, 5 mM GSH was added to the final mixture. The chemical shift perturbations that were observed comparing the ^1^H–^15^N HSQC maps before and after the addition of 5 mM GSH exclusively monitored the interaction of GSH with the ^15^N-labelled protein in the final mixture. The ^1^H–^15^N HSQC spectrum of the final mixture (i.e., with 5 mM GSH) was also compared with the ^1^H–^15^N HSQC spectrum of the starting ^15^N-labelled protein containing 5 mM GSH. The ^1^H–^15^N HSQC spectra showed 1) chemical shift changes exclusively due to the protein–protein interaction in the absence of GSH, and 2) further chemical shift changes upon 5 mM GSH addition. Moreover, when we compared the ^1^H−^15^N HSQC NMR spectrum of the final mixture (i.e., with 5 mM GSH) with the ^1^H–^15^N HSQC spectrum of the starting ^15^N-labelled protein containing 5 mM GSH, we found that they did not overlap. Collectively, all the NMR data indicated the formation of an apo hetero-complex in the absence of GSH and that GSH, which binds to the GLRX5 subunit of the hetero-complex, interacts at the protein–protein interface without promoting the releasing of the free proteins. The data reported in Figure 3 and Figure 4 were obtained through comparing the NMR spectra of the final mixtures with the starting ^15^N-labelled proteins, all containing 5 mM DTT and 5 mM GSH. These data thus included chemical shift changes due to two contributions, i.e., protein–protein and GSH–hetero-complex interactions.

### 4.4. Spectroscopic Methods to Characterize Fe–S Cluster-Binding and Transfer

All the spectroscopic experiments were performed under anaerobic conditions, preparing the samples in an anaerobic chamber (O_2_ < 1 ppm) and using degassed buffers, gas-tight cuvette, and gas-tight NMR tubes.

UV–VIS and UV–VIS CD spectra were performed at room temperature in 150 mM NaCl, 5 mM GSH, 5 mM DTT, and 50 mM phosphate buffer (pH 7.0) on a Cary 50 Eclipse spectrophotometer and JASCO J-810 spectropolarimeter, respectively.

Fluorescence emission measurements were performed at 298 K in 150 mM NaCl, 5 mM GSH, 5 mM DTT, and 50 mM phosphate buffer (pH 7.0) on a Cary 50 Eclipse spectrophotometer supplied with a single-cell Peltier thermostatic cell holder. After excitation at 280 nm, an emission scan was recorded between 200 and 600 nm. The emission spectra were corrected for the buffer baseline.

Paramagnetic 1D ^1^H NMR experiments were performed at 400 MHz with a ^1^H optimized 5 mm probe at 298 K in 150 mM NaCl, 5 mM GSH, 5 mM DTT, 50 mM phosphate buffer (pH 7.0), and 99% (*v*/*v*) D_2_O. These spectra were acquired by means of the super-WEFT sequence with a recycle time of 65 ms [62].

^1^H−^15^N HSQC NMR experiments were performed to follow the formation of [4Fe–4S]^2+^ NFU1 by mixing unlabeled [2Fe–2S]^2+^ C59Y BOLA3–GLRX5 and apo ^15^N-labelled NFU1 at a 1:1 ratio in the presence of 5 mM DTT and 5 mM GSH using the same experimental conditions previously applied to the wild-type hetero-complex [26].

### 4.5. Data-Driven Biomolecular Docking

Structural models of the apo BOLA3–GLRX5 and apo C59Y BOLA3–GLRX5 complexes were calculated using the protein–protein docking program HADDOCK 2.2 by following the standard HADDOCK procedure and employing the HADDOCK2.2 Web Server (https://alcazar.science.uu.nl/services/HADDOCK2.2/) [34,63]. Specifically, the structural models of the apo hetero-dimers were built from the structures of individual proteins (monomeric apo GLRX5 with a bound GSH molecule obtained from PDB entry 2WUL [64] and BOLA3 from PDB entry 2NCL [23]). The structure of the C59Y BOLA3 was obtained with Modeller 9.20 (https://salilab.org/modeller/) [65], using as template the existing NMR solution structure of BOLA3. The C59Y BOLA3 structure obtained by Modeller 9.20 was then energy minimized in explicit water using an AMBER 12.0 molecular dynamics program [66]. The NMR chemical shift mapping data obtained for the two hetero-complexes in the titrations performed at 800 μM starting protein concentration were used to define ambiguous interaction restraints for the residues at the interface. The “active” residues were defined as those having a chemical shift perturbation upon hetero-complex formation larger than the average of Δδ_avg_(HN) plus 1σ, as well as those residues whose backbone NHs broaden beyond detection upon the interaction with the protein partner, and all having a solvent accessibility higher than 50%; the “passive” residues were defined as those being surface neighbors to the active residues and having a solvent accessibility higher than 50%. NACCESS is the program used to calculate the atomic and residue accessibilities from the PDB file. GSH molecule was also included as active residue on the basis of the NMR data described in Section 4.3. The ensemble of 200 solutions was analyzed and clustered on the basis of the pairwise RMSD matrix calculated over the backbone atoms of the interface residues of GLRX5 after fitting on the interface residues of BOLA3 or C59Y BOLA3. This way of calculating RMSD values in HADDOCK resulted in high values that emphasized the differences between docking solutions. For this reason, we performed clustering using a 7.5 Å cut-off. The water-refined models were clustered on the basis of the default fraction of common contacts, FCC = 0.75, with the minimum number of elements in a cluster of 4. The clusters were ranked on the basis of the averaged HADDOCK score of their top 10 members and plotted against RMSD from lowest energy structure (Appendix A, Appendix A).

## Figures and Tables

**Figure 1 ijms-22-04848-f001:**
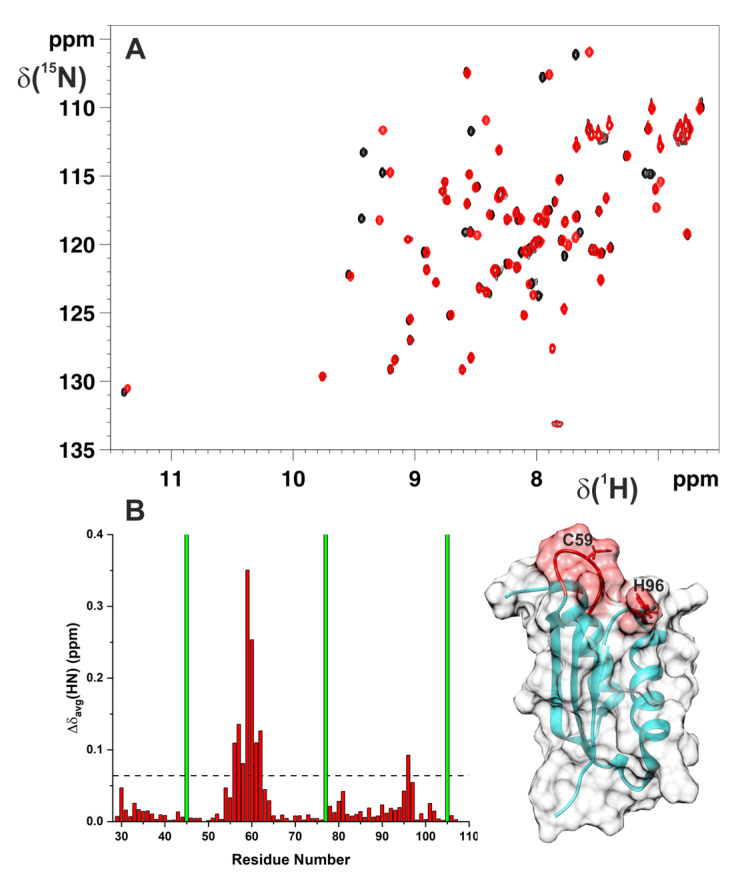
C59Y mutation structurally perturbed only the Fe–S cluster-binding region of BOLA3. (**A**) Overlay of the ^1^H−^15^N HSQC spectra of ^15^N-labelled BOLA3 (black) and ^15^N-labelled C59Y BOLA3 (red). The ^1^H−^15^N HSQC spectra were acquired at 298 K in 150 mM NaCl, 5 mM GSH, 5 mM DTT, 50 mM phosphate buffer (pH 7.0), and 10% (*v/v*) D_2_O. (**B**) Backbone weighted average chemical shift differences (Δδ_avg_(HN)) between BOLA3 and C59Y BOLA3. A chemical shift threshold value of 0.06, indicated as a dashed line, was estimated to define meaningful chemical shift differences (see the Materials and Methods section). The green bars indicate proline residues or unassigned NHs. The meaningful chemical shift differences are mapped in red on the solution structure of BOLA3 (PDB 2NCL). The side-chains of Cys59 and His96 Fe–S cluster ligands are depicted in red sticks.

**Figure 2 ijms-22-04848-f002:**
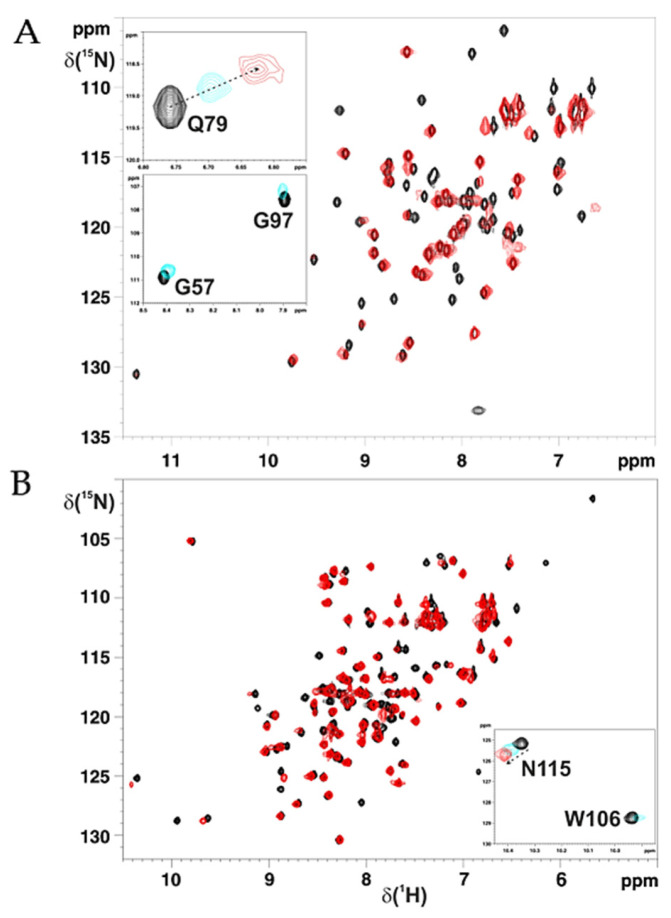
NMR showed apo hetero-complex formation between C59Y BOLA3 and GLRX5. (**A**) Overlay of the ^1^H−^15^N HSQC spectra of ^15^N-labelled C59Y BOLA3 (starting protein concentration 800 μM, black) and of a 1:1 ^15^N-labelled C59Y BOLA3-unlabelled apo GLRX5 mixture (red). In the insets, the overlay of ^1^H−^15^N HSQC spectra of ^15^N-labelled C59Y BOLA3 at 0.0 (black), 0.5 (cyano), and 1.0 (red) equivalents of unlabeled apo GLRX5 is reported, showing backbone NHs of residues in fast (Q79) and intermediate (G57 and G97) exchange regimes relative to the NMR time scale. (**B**) Overlay of the ^1^H−^15^N HSQC spectra of ^15^N-labelled apo GLRX5 (starting protein concentration 800 μM, black) and of a 1:1 ^15^N-labelled apo GLRX5-unlabelled C59Y BOLA3 mixture (red). In the inset, the overlay of ^1^H−^15^N HSQC spectra of ^15^N-labelled apo GLRX5 at 0.0 (black), 0.5 (cyano), and 1.0 (red) equivalents of unlabeled apo C59Y BOLA3 is reported, showing backbone NH of N115 in a fast exchange regime relative to the NMR time scale and the NH of the indole ring of W106 in an intermediate exchange regime relative to the NMR time scale. The ^1^H−^15^N HSQC spectra were acquired at 298 K in 150 mM NaCl, 5 mM GSH, 5 mM DTT, 50 mM phosphate buffer (pH 7.0), and 10% (*v*/*v*) D_2_O.

**Figure 3 ijms-22-04848-f003:**
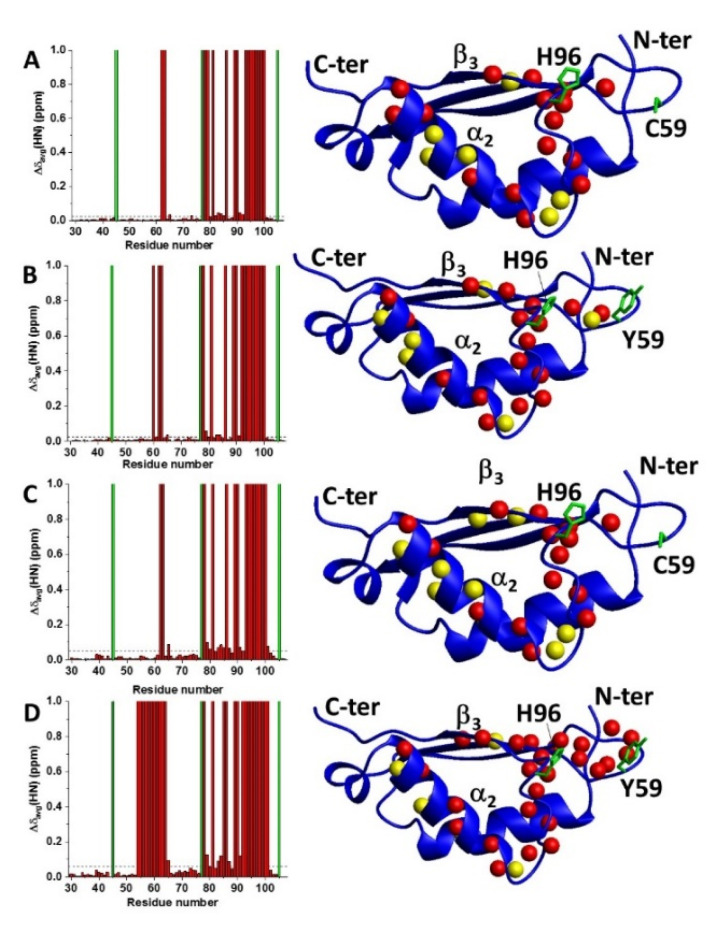
C59Y mutation promoted the formation of an aberrant protein–protein interaction. (**A**) Backbone weighted average chemical shift differences (Δδ_avg_(HN)) on ^15^N-labelled BOLA3 (100 μM starting protein concentration) upon hetero-complex formation with apo GLRX5 (left panel). Meaningful chemical shift differences are mapped on the solution structure of BOLA3 (PDB 2NCL) (right panel). (**B**) Δδ_avg_(HN) on ^15^N-labelled C59Y BOLA3 (100 μM starting protein concentration) upon hetero-complex formation with apo GLRX5 (left panel). Meaningful chemical shift differences are mapped on the structural model of C59Y BOLA3 (right panel). (**C**) Δδ_avg_(HN) on ^15^N-labelled BOLA3 (800 μM starting protein concentration) upon hetero-complex formation with apo GLRX5 (left panel). Meaningful chemical shift differences are mapped on the solution structure of BOLA3 (PDB 2NCL) (right panel). (**D**) Δδ_avg_(HN) on ^15^N-labelled C59Y BOLA3 (800 μM starting protein concentration) upon hetero-complex formation with apo GLRX5 (left panel). Meaningful chemical shift differences are mapped on the structural model of C59Y BOLA3 (right panel). Red bars with Δδ_avg_(HN) = 1 indicate residues whose backbone NH signals broaden beyond detection. The green bars indicate proline residues or unassigned NHs. Chemical shift threshold values (see the Materials and Methods section) are indicated as dashed lines. On the blue ribbon diagrams, backbone NHs showing line broadening beyond detection are depicted as red spheres, while backbone NHs with chemical shifts above the thresholds are depicted as yellow spheres. Sidechains of Cys59, Tyr59, and His96 are shown as green sticks.

**Figure 4 ijms-22-04848-f004:**
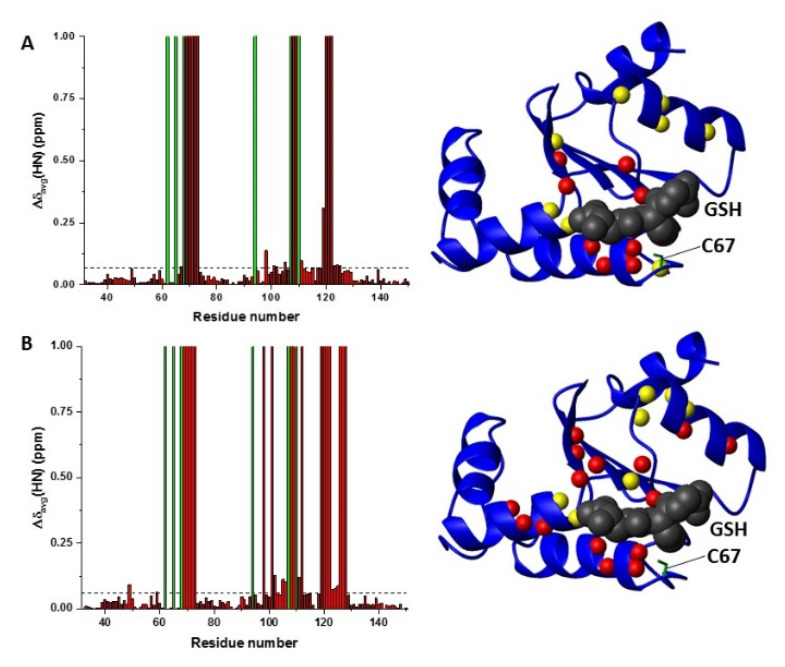
Mapping NMR chemical shift changes on GLRX5 upon apo hetero-complex formation. (**A**) Backbone weighted average chemical shift differences (Δδ_avg_(HN)) on ^15^N-labelled GLRX5 (800 μM starting protein concentration) upon complex formation with BOLA3 (left panel). Meaningful chemical shift differences were mapped on the solution structure of GSH-bound GLRX5 (PDB 2WUL) (right panel). (**B**) Δδ_avg_(HN) on ^15^N-labelled GLRX5 (800 μM starting protein concentration) upon hetero-complex formation with C59Y BOLA3 (left panel). Meaningful chemical shift differences were mapped on the solution structure of GSH-bound GLRX5 (PDB 2WUL) (right panel). Red bars with Δδ_avg_(HN) = 1 indicate residues whose backbone NH signals broaden beyond detection. The green bars indicate proline residues or unassigned NHs. Chemical shift threshold values (see the Materials and Methods section) are indicated as dashed lines. On the blue ribbon diagrams, backbone NHs showing line broadening beyond detection are depicted as red spheres, and the backbone NHs with chemical shifts above the thresholds are depicted as yellow spheres. The sidechain of Cys67 is depicted as green stick, and the GSH molecule is depicted as CPK mode in gray.

**Figure 5 ijms-22-04848-f005:**
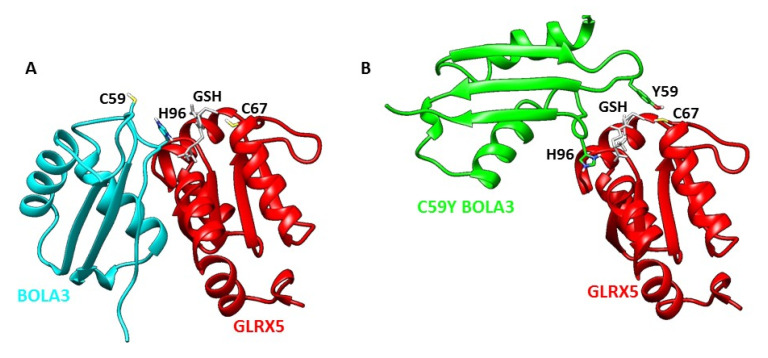
Visualizing the structure of the aberrant protein–protein complex by data-driven biomolecular docking. Structural models of apo BOLA3–GLRX5 (**A**) and C59Y BOLA3–GLRX5 (**B**) complexes. Ribbon diagrams of GLRX5, BOLA3, and C59Y BOLA3 structures are in red, cyan, and green, respectively. Sidechains of His96 and Cys59 in BOLA3, His96 and Tyr59 in C59Y BOLA3, Cys67 in GLRX5, and the GSH molecule bound to GLRX5 are shown as sticks.

**Figure 6 ijms-22-04848-f006:**
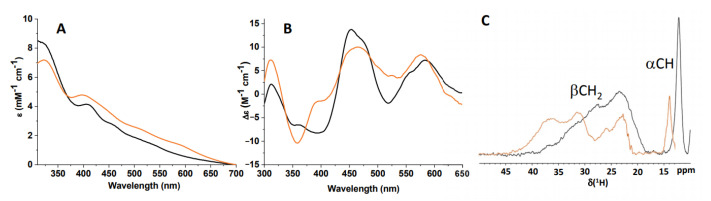
C59Y BOLA3–GLRX5 complex bound a [2Fe–2S]^2+^ cluster. UV–VIS (**A**), UV–VIS CD (**B**), and paramagnetic 1D ^1^H NMR spectra (**C**) of chemically reconstituted C59Y BOLA3–GLRX5 (black) and BOLA3–GLRX5 (orange) apo complexes.

**Figure 7 ijms-22-04848-f007:**
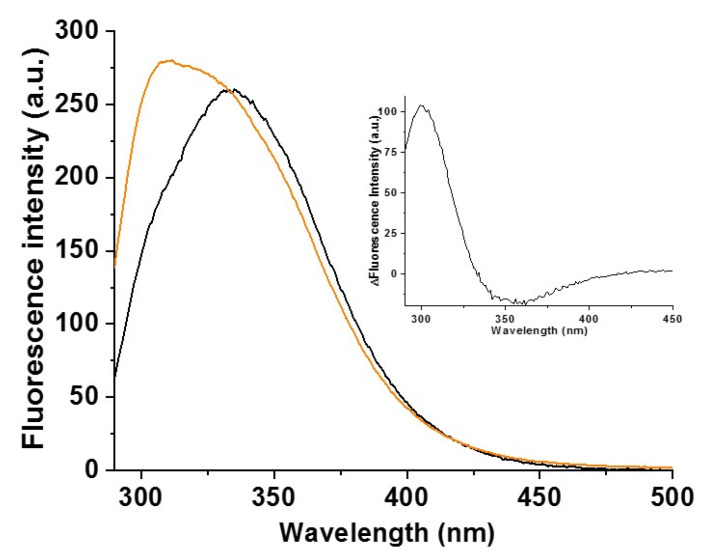
Fluorescence emission spectroscopy indicated that tyrosine 59 was not involved in cluster coordination in the C59Y BOLA3–GLRX5 complex. Fluorescence emission spectra of apo C59Y BOLA3–GLRX5 complex before (black) and after (orange) chemical reconstitution recorded upon excitation at 280 nm. In the inset, fluorescence emission difference spectrum (Δ) obtained by subtracting the spectrum of the chemically reconstituted hetero-complex from that of the apo hetero-complex.

## Data Availability

The data presented in this study are available within the article text, figures, and Appendix A.

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
