# Peer review of "Molecular Basis of Multiple Mitochondrial Dysfunctions Syndrome 2 Caused by CYS59TYR BOLA3 Mutation"

_ijms, 2021, doi:10.3390/ijms22094848_

Round 1
Reviewer 1 Report
In the manuscript “Molecular basis of multiple mitochondrial dysfunctions syndrome 2 caused by Cys59Tyr BOLA3 mutation “ by Giovanni Saudino et al. the authors describe the effect of a novel heterozygous variant in BOLA3 in which a paternally inherited mutation c.136C>T (p.Arg46*) previously reported is associated to a novel variant c.176G>A, p.Cys59Tyr maternally inherited. This particular biallelic variant is characterized by a mild phenotype of MMDS2 syndrome with a complete clinical recovery.
To investigate the effect of Cys 59 to Tyr mutation the authors analyzed the structure of BOLA3 protein using NMR and UV-Vis spectroscopy. The results reported indicated that the mutation does not affect the overall structure of the BOLA3 protein but affects its interactions with the partner protein GLRX5. Moreover, the aberrant interaction of the mutated BOLA3 protein with GLRX5 does not affects the affinity constant for the formation of the BOLA3-GLRX5 complex and maintain the ability of the complex to bind a [2Fe-2S]2+ cluster.
The authors conclude that the key role of the C59T mutation in the development of the MMDS2 syndrome could be ascribed to two main factors: an increased folding instability of the protein and the formation of an aberrant complex with the GLRX5 partner protein that can alter its ability to transfer [2Fe-2S]2+ clusters.
In this contest, the pathological effect of this mutation is more evident in the early stage of life when the energetic demand is higher and is less evident later when the energetic demand is lower explaining the complete clinical recovery of the patient.
The structural data presented appear to be sound and the manuscript is clearly written. A limitation of the work is the lack of data on the ability of the aberrant complex BOLA3-GLRX5 to transfer iron Sulphur clusters. It would have been particularly interesting to know if the rate of transfer of the [2Fe-2S]2+ clusters for example to human apo ferredoxin is slower with respect the wild-type complex.
Reviewer 2 Report
The paper entitled “Molecular basis of multiple mitochondrial dysfunctions syndrome 2 caused by Cys59Tyr BOLA3 mutation” by Saudino et al reports on the impact of the Cys59Tyr mutation on human BOLA3 found in a recently described human patient (i) for the formation of a complex between BOLA3 and the GLRX5 partner protein, and (ii) for the Fe-S cluster binding properties of the complex. The results show that tyrosine 59 cannot serve as an Fe-S cluster ligand unlike the cysteine but that the [2Fe-2S] cluster usually bound by the heterocomplex still forms despite the mutation structurally perturbs the Fe-S cluster binding region as evidenced by NMR studies. Also, the mutation was found to promote the formation of an aberrant apo C59Y BOLA3-GLRX5 complex. It is proposed that the formation of this aberrant apo-complex represents the cause of the pathogenic effects induced by the Cys59Tyr mutation.
The study is well designed and the results usually adequately described. My main concern here is that the functional unit is supposed to be the holo-heterocomplex, suggesting that the defects associated with the mutation may rather originate from a defect in the Fe-S cluster transfer to recipient proteins. Hence to strengthen the conclusions of this study and since the authors have previously worked on the Fe-S cluster transfer from GLRX5-BOLA3 to NFU1, I would recommend to compare the rate or efficiency of the transfer using the complex containing the mutated BOLA3. This would help clarifying the effect of the mutation.
A second comment is related to the design of the NMR experiments for which 5 mM GSH and DTT has been added. This may require a better justification. Unless I missed something, it is not clear to me whether we can discriminate the effect of GSH on GLRX5 from the effect of BOLA3? Also, is there an effect of GSH on BOLA3?
A last comment is related to the conclusion that the Cys59Tyr mutation does not significantly affect the affinity constant relative to complex formation with GLRX5 (mentioned in both the result and discussion parts).
This comes from an NMR titration where a 1:1 15N-labelled BOLA3-15N-labelled C59Y BOLA3 mixture was titrated with unlabeled apo GLRX5 (in two fold excess). I am unsure that the fact that the added apo GLRX5 was equally partitioned between the two BOLA3 proteins is sufficient to make this statement. Can we exclude that the formation of the complex involving the C59Y BOLA3 occurs after the regular heterocomplex is formed ? For this reason, I find overstated to speak about affinity constant, which should in fact be assessed by other specific experiments.
Minor comments:
L44: define MRI.
L65: replace by “was shown not to be required…”
At several places in the text (l159 and beyond), the units do not appear correctly.
L348-349: There should a problem with the end of the following sentence, please rephrase. “We showed that the Cys59Tyr mutation of BOLA3 does not have severe effects on the protein structure and folding, which resulted indeed preserved.”
Reviewer 3 Report
Saudino et al described the molecular basis of the Cys59Tyr BOLA3 pathogenic mutation in MMDS2 to decipher this MMD2’s phenotype never described and documented before.
The authors reported with the clear support of experimental evidence that the mutation does not modify the 3D structure of BOLA3, but it only perturbs the Fe-S cluster binding region, that this mutation causes the formation of an aberrant apo C59Y BOLA3-GLRX5 complex that is clearly different from the one formed by the wild-type proteins, and, finally, that it determines a perturbation in the chemical environment of the iron sulfur cluster.
This manuscript clearly also shows a diverse set of complexes analyzed successfully by NMR, UV-visible and protein-protein docking. All the models are well supported by experimental data.
This manuscript is well written, clear, and the conclusions are supported by all the experimental evidence. Every steps of the experiments are diligently explained.
Minor points:
In the introduction the sentence “All these five genes…” can be re-written in a clearer way? As far as I understood the delivery of [4Fe-4S] cluster to mitochondrial protein relies on these five genes but it seems that these five genes are involved only for aconitase and lipoic acid. Is it correct or these genes deliver the iron sulfur cluster also to other proteins? Can the authors explain better?
The Greek letter in the manuscript and in the captions are not visible correctly (they are something like @), please correct this unless this is an editorial issue.
The model generated with MODELLER have been minimized in AMBER in order to relax the system after the modelling and to see how and if the Y59 reorient in a different position? Can the authors comment on this eventually?
Can the authors comments also on the surface change induced on the BOLA3 by the tyrosine and how this change might impact the binding of the cluster and/or binding of GLRX5?
Do the authors tried to perform a docking in presence of two molecules of GSH once the Y59 is present? It might be speculative but is there space for a second GSH molecule to bind the cluster? Is there any models with iron-sulfur cluster bound available?
I recommend for acceptance after minor revision
